# Clot Morphology in Acute Ischemic Stroke Decision Making

**DOI:** 10.3390/ijms232012373

**Published:** 2022-10-15

**Authors:** Joanna C. Huang, Sonu M. M. Bhaskar

**Affiliations:** 1Global Health Neurology Lab, Sydney, NSW 2000, Australia; 2Neurovascular Imaging Laboratory, Ingham Institute for Applied Medical Research, Clinical Sciences Stream, Sydney, NSW 2170, Australia; 3UNSW Medicine and Health, University of New South Wales (UNSW), South Western Sydney Clinical Campuses, Sydney, NSW 2170, Australia; 4Department of Neurology & Neurophysiology, Liverpool Hospital & South West Sydney Local Health District (SWSLHD), Sydney, NSW 2170, Australia; 5Stroke & Neurology Research Group, Ingham Institute for Applied Medical Research, Sydney, NSW 2170, Australia; 6NSW Brain Clot Bank, NSW Health Pathology, Sydney, NSW 2170, Australia

**Keywords:** stroke, thrombosis, biomarkers, diagnosis, cryptogenic, histopathology, thrombectomy

## Abstract

Stroke is a leading cause of death and disability in the world, and the provision of reperfusion therapy and endovascular therapy, in particular, have revolutionized the treatment of patients with stroke and opened opportunities to look at brain clots retrieved after the procedure. The use of histopathology and molecular profiling of clots is of growing research and clinical interest. However, its clinical implications and incorporation within stroke workflows remain suboptimal. Recent studies have indicated that the study of brain clots may inform the mechanism of stroke and hence guide treatment decision-making in select groups of patients, especially patients without a defined cause or known mechanism. This article provides a comprehensive overview of various clot histopathological examinations in acute stroke-care settings, their clinical utility, and existing gaps and opportunities for further research. We also provide targeted recommendations to improve clot analysis workflow, hence standardizing its incorporation into clinical practice.

## 1. Introduction

Stroke is classically defined by the World Health Organization as “rapidly developed clinical signs of focal (or global) disturbance of cerebral function, lasting more than 24 h or leading to death, with no apparent cause other than of vascular origin” [1]. In 2019, stroke was the second leading cause of death worldwide [2], and the third leading cause of disability [3]. Acute ischemic stroke (AIS) accounts for approximately 80% of this burden, most commonly caused by thromboembolism, arising from large artery atherosclerosis (LAA), or of cardioembolic origin such as in atrial fibrillation [4]. However, in 25–40% of AIS patients, the underlying cause of stroke remains cryptogenic [5]. Incorporating clot histopathological analyses in the etiology workup may improve the reclassification of cases that would otherwise be presumed to be cryptogenic [6,7].

The opportunity to investigate clot histopathology was limited prior to 2015, when the publication of five randomized controlled trials [8,9,10,11,12] led to endovascular thrombectomy (EVT) becoming the standard of care. Now, clots are routinely extracted from patients in clinical practice, allowing detailed histopathological and immunohistochemical analyses of stroke clots [6], which could pave the way to optimizing interventional and clinical decisions and outcomes [13]. Whilst qualitative assessment of the embolus is still crucial in the diagnosis of individual cases, especially when identifying atypical causes of stroke, there is as yet no definitive clot signature for the precise diagnosis of the etiology of stroke [14,15,16]. A recent meta-analysis from our group found that the fibrin composition is significantly higher in strokes of cardioembolic and cryptogenic origin, and RBC content is favorably correlated with the hyperdense middle cerebral artery sign (HMCAS) and better reperfusion results [17].

Currently, the AIS workup includes neuroimaging, typically with non-contrast computed tomography (NCCT) or magnetic resonance imaging (MRI), along with CT angiography, CT perfusion, and other advanced imaging techniques, in order to exclude hemorrhagic stroke and guide intervention decisions [18]. Intravenous thrombolysis (IVT) and EVT are the two central management strategies, both of which aim to rapidly save as much brain tissue as possible [4]. IVT with recombinant tissue plasminogen activator (rt-PA) can only be delivered to eligible patients within 4.5 h of stroke onset [19,20]. Less than 15% of AIS patients receive IVT [21,22], with some studies reporting as low as 3.8% [23]. EVT involves the mechanical removal of the clot via the femoral artery, with several procedural technique options (Figure 1). Approximately 9% of patients are eligible, and even fewer undergo the procedure [24,25]. Considering these poor treatment rates, improvements must be made in stroke management options and application. 

This scoping review presents an overview of the current knowledge about clot histopathological features, and their relation to stroke etiology, outcomes, imaging, and management. 

## 2. Pathogenesis of Thromboembolism 

The process of thrombus formation is initiated by damage to the endothelium, such as in atherosclerosis or following myocardial infarction. Exposed collagen causes platelets to change shape and adhere to the vessel wall. Activated platelets and von Willebrand Factor (VWF) help promote further platelet aggregation [28]. Immune mediators are also involved, with neutrophils releasing neutrophil extracellular traps (NETs), and monocytes aiding red blood cell (RBC) recruitment and fibrin formation (Figure 2) [29]. The key components of a thrombus, therefore, include RBCs, white blood cells (WBCs), platelets, fibrin, VWF, and extracellular DNA [30]. 

An overview of these components will be discussed, followed by a critical evaluation of the evidence regarding the association of clot features with stroke etiology, imaging, management, and prognosis, all of which are summarized in Table 1. The various histological stains and immunohistochemical analyses investigated are summarized in Table 2. 

## 3. Clot Components

Various clot components and their clinical significance are discussed below. 

### 3.1. Red Blood Cells

RBCs are a major component of thrombi and had been perceived to play a passive role in clot formation, but recent research has revealed direct rheological mechanisms contributing to thrombosis as well as interactions with platelets, endothelium, thrombin, and plasmin [30,73,74,75]. Under histological examination, RBC-rich clots are composed of packed RBCs within a thin meshwork of fibrin [76]. The thin fibrin layers make the clot more susceptible to rt-PA thrombolysis [50,53,77]. These clots are also more successful with EVT [41,44,45], as they have less friction [78], are less stiff [79], and have better stent integration or conformability into an aspiration catheter [77]. Several studies have reported that RBC-rich clots are associated with a non-cardioembolic cause, often LAA [32,34,35,40,44]. Significant radiological features of RBC-rich clots are: HMCAS [39,40,42,44,50,54] and susceptibility vessel sign (SVS) [37,53,80]. 

### 3.2. Polyhedrocytes 

Polyhedrocytes form when activated contractile platelets pull on fibrin, causing compression of RBCs [76]. They form a nearly impermeable seal, hindering the effective diffusion of fibrinolytic enzymes into thrombi. Khismatullin et al. [66] associated polyhedrocytes with the decreased success of IVT and higher National Institutes of Health Stroke Scale (NIHSS) scores. Very few studies describe polyhedrocytes in stroke clots and more data are warranted. 

### 3.3. Platelets/Fibrin

Platelet-fibrin (PF) rich clots are more complex than RBC-rich areas, containing VWF [60], WBCs, NETs, and extracellular DNA [76]. These non-fibrin components have been described to form an outer ‘shell’ that impedes rt-PA thrombolysis as there is more mechanical stabilization [77]. Furthermore, plasminogen activator inhibitor 1 (PAI-1), and protease nexin-1 (PN-1) [46] which inhibit the action of rt-PA were found within this shell. PF-rich clots are stiffer, thereby harboring a stronger interaction with the vessel wall [33]. These mechanisms explain the findings that PF-rich clots are more resistant to both IVT and EVT [35,60]. Many studies find that PF-rich clots are more abundant in cardiogenic pathology [32,33,34,35,36,49]. 

### 3.4. White Blood Cells

The role of WBCs in thrombosis is being expanded upon in recent research, playing a part in releasing proinflammatory and procoagulant mediators [31,81], but also thrombus resolution by modulating fibrinolysis pathways to restore patency of flow [81]. Higher WBC proportions correspond with worse recanalization outcomes [42], and with cardioembolic origin [34,35,36,42,82]. These thrombi are associated with higher NIHSS scores [31]. Higher neutrophil and lymphocyte content also corresponds to a higher incidence of clot fragmentation [36,70].

### 3.5. Neutrophil Extracellular Traps 

Neutrophil Extracellular Traps (NETs) are formed when activated neutrophils release DNA as decondensed chromatin, histones, proteolytic enzymes, and neutrophil granule proteins, such as MPO and neutrophil elastase [33,83]. NETs immobilize large pathogens and reduce infection, but also operate as a scaffold for RBCs and platelets, effectively promoting thrombus formation and coagulation [33,84]. Furthermore, the components of NETs may also limit the action of ADAMTS13, the enzyme which cleaves VWF [85]. Higher NET content corresponds to worse recanalization outcomes [68], more thrombectomy attempts [63,64,68,77], higher NIHSS score at discharge [64], and cardioembolic origin [63]. Recent research has also implicated a pathological role of NETs in ischemic stroke indicating the therapeutic potential of NET-inhibitory factor (nNIF) in stroke [86,87]. A study by Denorme et al. reported NET-forming neutrophils throughout the brain tissue of patients who had ischemic strokes [86]. The authors also found that increased plasma NET biomarkers were associated with worse stroke outcomes. Furthermore, stroke patients had elevated plasma and platelet surface-expressed high-mobility group box 1 (HMGB1). Another study by Dhanesha et al. found that the nuclear pyruvate kinase muscle 2 (PKM2), a regulator of systemic inflammation, was upregulated in neutrophils following the onset of ischemic stroke in both humans and mice [87]. Neutrophil hyperactivation was mediated by PKM2. By reducing postischemic cerebral thrombo-inflammation, PKM2 deficiency in myeloid cells improved both the short- and long-term outcomes of stroke [87].

### 3.6. Von Willebrand Factor 

VWF is a large glycoprotein produced by megakaryocytes and endothelial cells. It plays an essential role in hemostasis and thrombosis by promoting platelet adhesion and supporting the action of important proteins involved in coagulation such as Factor VIII [22]. A higher VWF content is associated with increased rt-PA resistance, possibly due to its tight interaction with fibrin, and is linked with a higher NIHSS score [57]. 

### 3.7. Extracellular DNA

Extracellular DNA is mainly found on the interface between PF-rich and RBC-rich areas, alongside WBCs [21,47]. Extracellular DNA and histones can modify the structure of fibrin, with Staessens et al. [47] speculating that it likely impedes responsiveness to IVT. DNA is typically higher in cardioembolic thrombi [69].

### 3.8. Endothelial Cells 

Endothelial cells have also been observed, indicating a potential role in thrombus formation via endothelialization over and within the thrombus [33], as well as interaction with immune cells [38]. Endothelial cells theoretically limit rt-PA penetration into thrombi, but Almekhlaf et al. [48] noted that in the three thrombi they observed with this phenomenon, the degree of endothelialization would not affect dissolution. 

### 3.9. Bacteria

The presence of bacteria within clots can indicate infectious stroke etiologies such as septic emboli from infective endocarditis, though infections generally have been associated with a higher risk of thrombosis and AIS [22]. One study identified paucicellular fibrinoid material as a potential marker for septic emboli [7]. Clot bacterial content is associated with worse recanalization outcomes and increased thrombectomy attempts [43], though further research is necessary to draw stronger conclusions. 

### 3.10. Calcifications 

Calcified thrombi are uncommonly extracted and studied in stroke thrombectomy, despite the high prevalence of atherosclerotic plaque calcification. Potential sources include calcified aortic stenosis, carotid atherosclerosis, and mitral annular calcification, on top of cardiovascular procedural causes [70,71]. Calcium increases rt-PA resistance and thrombectomy attempts [48,70,88], likely due to increased stiffness. On imaging, the calcific emboli appear more hyperdense and rounder than non-calcified clots [71]. Recurrent stroke is common in this patient group, and they experience worse clinical outcomes with higher mortality rates [72]. 

### 3.11. Other 

Other rare components that have been identified within stroke thrombi include cholesterol, collagen, vascular wall components, myxomatous material, and non-infective vegetations [21,52,89,90]. 

## 4. Clot Composition and Etiology 

Delineation of stroke etiology is important in guiding the appropriate treatment strategy for both acute management and secondary stroke prevention. Most studies, including one of the largest by Sporns et al. (*n* = 187) [35] found an association between RBC-rich clots and LAA, and between PF-rich clots and cardioembolism [32,34,35,40,44,49]. However, Kim et al. [37] reported the opposite, though their patient cohort was smaller with only eight clots of LAA origin. Fitzgerald et al. [56] found an association between platelets and LAA source in a study with 105 patients, though they considered platelets alone, while other studies grouped platelets and fibrin together. In addition, a meta-analysis by Brinjikji et al. [80] in 2015 found no significant relationship between composition and etiology, though it only analyzed the hematoxylin and eosin staining results, and needs to be re-investigated incorporating recent studies. There is wide variability in findings, which limits the clinical applicability, and deeper investigation is warranted. Furthermore, WBCs [34,35,42], NETs [63], and DNA [69] seem to be more prevalent in cardioembolic clots, but more studies are necessary to strengthen this. VWF may be higher in the aortic arch and carotid artery calcifications [88]. Determining stroke cause is important to begin prompt treatment to prevent secondary stroke, such as anticoagulation for atrial fibrillation, antiplatelet therapy for LAA, or antibiotics for septic causes. Notably, several studies also support the finding that most cryptogenic strokes may be reclassified or attributed to cardioembolic etiology [34,35,36]. Patent foramen ovale (PFO)-induced paradoxical embolism is likely the most common cause of cryptogenic stroke [91]. A PFO may be presented in up to 30% of the population. Individuals with a PFO have a 0.1% and 1% annual risk of cryptogenic and recurrent strokes, respectively [92]. Interestingly, extracellular vesicles (EVs) derived from blood may be of potential utility in stroke etiology delineation and indicating the state of cerebrovascular disease [93]. EVs can mediate coagulation through anticoagulant or fibrinolytic pathways in addition to their procoagulant capabilities [94,95]. In this context, the evaluation of EVs in isolated thrombus and their putative role in differentiating stroke types offer another avenue for future research [95]. 

## 5. Clot Composition and Imaging 

Imaging is a crucial investigation before stroke management decisions can be made. Most studies concur that RBC-rich thrombi are associated with hyper-attenuation or the presence of the HMCAS and SVS, seen on CT and MRI, respectively [21,54,80]. Contrastingly, two small studies reported no significant association, though one was limited by a majority of its specimens being fragmented [32,65]. Moreover, whether these signs are clinically significant or not is contested. Various studies, including a systematic review, concluded that the HMCAS was associated with successful recanalization [80,96,97], though a smaller study by Ye et al. reported no significant association [39]. Their results may have been affected by the small sample size and limited by only studying patient cases with retrievable clot specimens. Thrombus perviousness can also be measured by comparing NCCT against CT angiography attenuation, indicating the degree of contrast penetration and thus permeability. Increased perviousness is widely associated with greater success in IVT and patient outcomes [55,98]. The largest study on this topic by Benson et al. [55] found clot perviousness to be positively correlated with RBC content. However, two other studies linked perviousness with PF-rich clots [49,67]. Patel et al. [67] suggested that the statistical significance of Benson et al. was limited by dichotomizing perviousness and clot composition rather than considering them as continuous variables. However, they also recognized that their study did not control for all variables. In all cases, neuroradiologist ability and error are also limitations to consider. Finally, the presence of calcification may also correspond to increased hyperdensity on imaging [71]. Overall, imaging markers have the potential to inform stroke workflows but the discrepancy between studies, and the limited knowledge, warrant more research in order to draw clinically significant conclusions. The resolution of imaging modalities to accurately measure HMCAS and SVS may depend on the size of clots/vessels being imaged and the thickness of the multiplanar imaging of the skull. Thin-section multiplanar imaging should be preferred [99].

## 6. Clot Composition and Treatment 

Several studies have found that IVT is more successful with RBC-rich clots, explained by the thin fibrin meshwork which it is easier for the rt-PA to dissolve [50,53,76,77]. Meanwhile, PF-rich clots have additional components involved which complicate thrombolysis. There is denser fibrin, VWF, extracellular DNA, and NETs, altogether decreasing the thrombus’ permeability, in particular acting as a ‘shell’ [46,60,68,85]. DNA and histones crosslink to and thus modify the structure of fibrin, resulting in thicker fibers that are harder to lyse [100]. These complex components also impair EVT, by making PF-rich clots ‘stiffer’ and thus less integrative, compared to the softer, lower friction RBC-rich clots [33,79]. In addition to PF-rich clots, polyhedrocytes, WBCs, calcifications, bacterial presence, and endothelial cells also predict unfavorable treatment outcomes [42,66]. 

No contradictory evidence was found; however, several studies have reported no significant association between composition and treatment success [32,37,38,56]. Thus, while the evidence seems promising, larger studies or meta-analyses are needed in order to confirm and solidify the clinical utility of these findings. Furthermore, studies are limited by the lack of a consistent method to evaluate how the EVT technique impacts treatment success, which could significantly affect the results. One study which does explore EVT techniques suggested that a direct aspiration first pass technique (ADAPT) benefits RBC-rich thrombi while balloon guide catheter (BGC) techniques suit PF-rich clots [21]. Strengthening knowledge about the relationship between clot composition and EVT devices could play an integral part in improving EVT efficacy. 

## 7. Clot Composition and Clinical Outcomes 

While successful recanalization is a therapeutic goal, for 30% of stroke patients who achieve complete reperfusion it is not conducive to good clinical outcomes [101]. Better outcomes have been associated with greater clot perviousness, shorter thrombi, and a shorter procedural time [97]. More severe strokes are associated with higher polyhedrocytes, platelets, fibrin, VWF, WBC, and NET content, as well as the cardioembolic cause [31,34,60,77]. Clot fragmentation is a frequent complication; however, whether RBC-rich or PF-rich clots are more prone to embolization remains unclear. RBC-rich clots could predict pre-interventional migration and peri-EVT fragmentation due to their softer nature [13,82]. Other studies find PF-rich clots more prone to secondary embolism [59]. Administering rt-PA has also been associated with pre-EVT distal embolization [58,102], which is an issue that affects clot inaccessibility [97]. Several factors may influence this contradiction, including the variation between studies in EVT methods. For example, one study employed the use of BGCs [58], whereas the others did not. This is significant as BGCs arrest the antegrade blood flow and can reduce distal embolization [103]. One study found that BCGs decrease embolization rates particularly for softer clots, and Solumbra is more suitable for harder clots [104]. Another study associated fragmentation with combining stent retriever and contact aspiration, as well as with a higher number of passes, higher lymphocyte count, and a longer procedural time [36]. 

## 8. Gaps and Limitations 

While many significant advancements have been made in clot histopathology, we have identified several gaps in this literature review. Selection bias is inevitable, as only patients with successful EVT could be studied. Irretrievable clots, clots that were rapidly dissolved by thrombolysis, or those ineligible for treatment are therefore underrepresented. Furthermore, several studies included patients regardless of rt-PA administration, or EVT technique, both of which could influence clot characteristics. One study found that stent retrievers ‘crush’ and reduce thrombus size [65]. Another potential limitation is the inconsistency in clot analysis and reporting, with differing stains and methods leading to unwarranted variations in clinical care. Therefore, clot histopathology protocols must be standardized across centers. Finally, despite the emerging body of evidence, translation into clinical practice and workflow improvements has been challenging, and more research is required to improve this aspect of stroke diagnostic workup. 

## 9. Future Directions 

### 9.1. Imaging and Thrombectomy Technique

Despite the correlation between the HMCAS or SVS with RBC-rich thrombi, the clinical utility of imaging markers is limited. Bridging this gap comes with determining which EVT methods are most suitable for RBC-rich vs PF-rich thrombi. Utilizing bioimpedance and electrical impedance spectroscopy to observe clot morphology could provide insight during procedures, alongside new-generation guidewires, sensors, and catheters [21]. One recent study using digital subtraction angiography imaging during EVT indicated that confirming stent integration into a clot could improve first-pass retrievals [105]. Finally, novel thrombectomy tools, such as the EmboTrap device which uses a distal tip for capturing clot fragments, may improve EVT outcomes [106]. 

### 9.2. Novel Therapeutic Targets

Given the small number of patients who can benefit from rt-PA thrombolysis, there is a compelling need for novel thrombolytic agents targeting non-fibrin clot components. Studies have shown that ADAMTS13 or DNAse-1 which cleave VWF and DNA, respectively, are more successful at dissolving rt-PA-resistant clots than rt-PA alone [22,31,82]. Thrombolytic drugs could also aim to target PAI-1 and thrombin-activatable fibrinolysis inhibitors (TAFI) [22]. Targeting the process of immuno-thrombosis with Cl-amidine, which inhibits protein arginine deiminase 4 (an enzyme promoting NET formation), has also been shown to improve stroke outcomes in an animal study [85]. Furthermore, anti-inflammatory drugs (e.g., roflumilast) can suppress platelet, neutrophil, and NET activation [31], and immunotherapy can target interactions between immune and coagulation pathways [107]. 

### 9.3. Machine Learning

Application of machine learning and computer vision platforms are an important avenue of future research that may automate the analysis and reporting of clot histopathology, hence expediting clinicopathological workflows. Employing algorithms in the process of stroke etiology delineation could aid clinicians in predicting stroke causes and minimizing cryptogenic stroke [104].

## 10. Conclusions

Clot immunohistopathology has divulged valuable information in recent years and is still developing. Several studies have reported that RBC-rich clots predict better reperfusion and patient outcomes, have an arteriogenic origin and are identifiable as hyperdense on imaging. Conversely, clots dense in platelets and fibrin tend to be comprised of more WBCs, VWF, and NETs which release DNA, complicating treatment success. The variability between studies remains a notable barrier to the clinical applicability of these results, which may pertain to factors such as inconsistencies in immunohistochemical analysis protocols, and small sample sizes. Future multi-center large-scale studies investigating clot composition and its implications on stroke causes and outcomes, with harmonized clot collection, processing, examination, and reporting protocol, may address these gaps and provide a means to develop evidence-based clinical indications or guidelines.

## Figures and Tables

**Figure 1 ijms-23-12373-f001:**
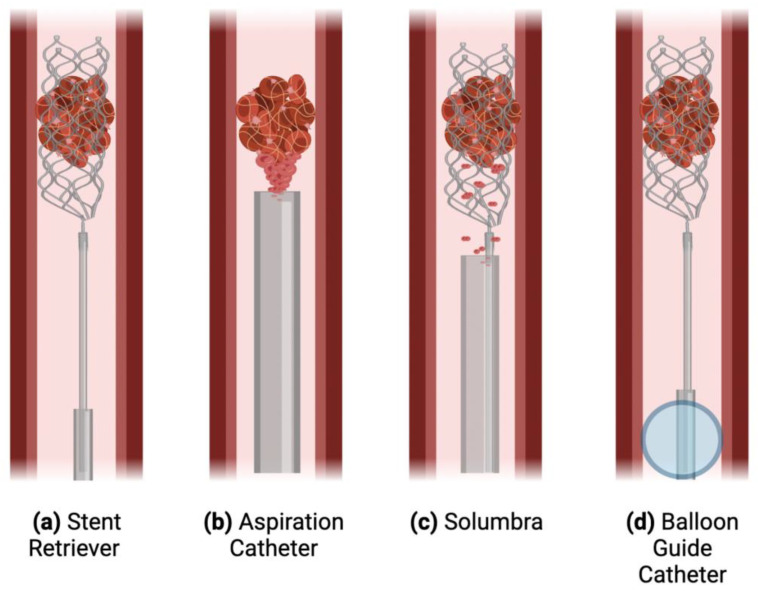
**Various techniques used in endovascular thrombectomy.** A schematic displaying four techniques: (**a**) The stent-retriever technique in which an aspiration catheter (e.g., a catheter with an inner diameter (ID) of 0.060” or 0.070”) is inserted through the femoral or radial artery and guided up to the specific vessel (e.g., approximately 2.5–3.0 mm in the M1 segment of the middle cerebral artery (MCA)) blocked by the clot. The guidewire and microcatheter cross the clot, and the stent is unsheathed and integrated with the clot, providing immediate reperfusion in most cases. Then, the stent and microcatheter are retrieved, ideally along with the clot [26]. (**b**) Aspiration catheters, which are larger than a microcatheter but not a fully deployed stent retriever, aspirate the clot in order to remove it from circulation [27]. (**c**) The Solumbra technique utilizes both the above techniques, simultaneously aspirating and using a stent to retrieve the clot. (**d**) A balloon guide catheter can be inflated in place in the vessel, stopping antegrade blood flow and causing brief retrograde flow, diminishing the risk of distal embolization. *Adapted from Munich et al. (2019)* [26].

**Figure 2 ijms-23-12373-f002:**
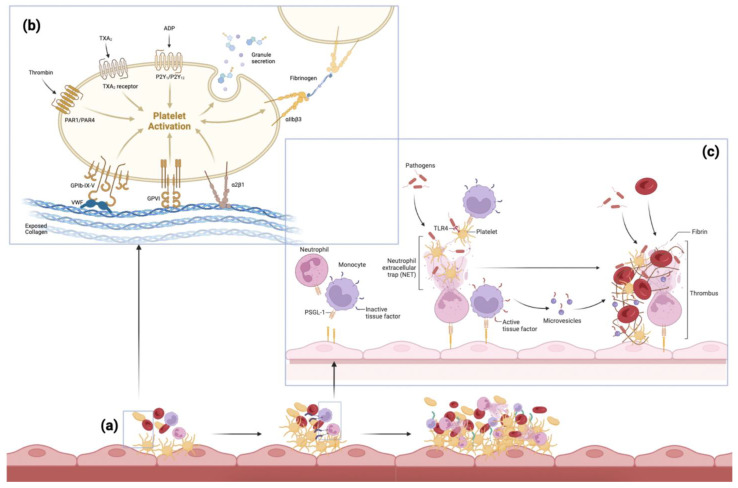
**The pathogenesis of thromboembolism.** Various pathways implicated in the pathogenesis of thromboembolism: (**a**) The accumulation of thrombotic mediators and cells, starting with primary hemostasis including platelet activation and immuno-thrombosis, followed by secondary hemostasis forming a thrombus. (**b**) The process of platelet activation: exposed connective tissue and collagen cause platelets to change shape and adhere to the vessel wall via pseudopods and platelet receptors. Activated platelets release factors from their granules such as adenosine diphosphate (ADP) (causing nearby platelets to adhere, creating a platelet plug), thromboxane A2 (TXA_2_) (promoting platelet aggregation and release of more ADP), and thrombin (a strong platelet agonist converting fibrinogen to fibrin). The von Willebrand Factor (VWF) helps platelets adhere via platelet receptor Glycoprotein Ib (GP Ib) on the platelet membrane [28]. (**c**) Neutrophils release neutrophil extracellular traps (NETs), which further promote thrombus formation and help platelet activation. Monocytes express tissue factors on their surface, and aid red blood cell (RBC) recruitment and fibrin formation [29]. *Adapted from Engelmann and Massberg (2013) and Bi et al. (2021)* [29,31]. Abbreviations: TXA_2_: thromboxane A2; ADP: adenosine diphosphate; PAR: protease-activated receptor; GP: glycoprotein; VWF: von Willebrand factor; NET: neutrophil extracellular trap; RBC: red blood cell; TLR: toll-like receptor; PSGL-1: P-selectin glycoprotein ligand-1.

**Table 1 ijms-23-12373-t001:** Clot histology stains and immunohistochemical analyses used by various studies: Uses, advantages, and disadvantages.

Histological Stain	Uses	Advantages	Disadvantages	Reference Studies
**Hematoxylin and Eosin**	Differentiate between cell nuclei and cytoplasm	Common Cheaper Effective	Can only identify 3 major constituents (RBCs, WBCs, and Fibrin/Other) Insufficient discrimination between fibrin and platelets	[7,32,33,34,35,36,37,38,39,40,41,42,43,44,45,46,47,48,49,50,51,52,53,54,55,56,57,58,59]
**Martius Scarlet Blue**	Anatomical details	Good color differentiation makes characterization more accurate Can identify platelets-rich regions/clots as distinct from fibrin [51]	More expensive Needs experienced histopathologist requiring more skill Stain outcome is dependent on the differentiation step	[33,39,45,47,48,51,53,55,56,60]
**Gram Stain**	Can look at bacterial colonization	Widely used Can assist in identifying infectious stroke etiology	Only investigates bacteria	[7,43]
**Elastica van Gieson**	Can visualize connective tissue—elastic fibers and collagen	Commonly used Widely available Inexpensive [61]	Easy to over-differentiate	[7,35,36,42,52,59]
**Prussian Blue**	Detects hemosiderin or iron [7] in macrophages (siderophages) or RBC lysis	Can help age the thrombus	Less utility overall	[7,35,36,59]
**Masson’s Trichrome**	Collagen, muscle, and bone	Can be useful in characterizing atherosclerotic plaques and when LAA origin is suspected	Poorly differentiates between clot components	[7,33,41,48,52]
**Ladewig’s Trichrome**	Fibrin deposits	Can also differentiate RBCs, muscle, nuclei, and cytoplasm	Less common?	[36]
**Feulgen’s Stain**	DNA	Highly sensitive for observing DNA and nuclei [62]	More complex staining procedure	[47]
**Von Kossa**	Detects mineralization	Confirms suspected calcification	Only observes mineralization	[7,33,36,48]
**Naphthol AS-D chloroacetate stain**	Granulocytes	Observes the presence of macrophages	Less common	[36]
**Mallory’s phosphotungstic acid hematoxylin **	Fibrin and Collagen	Can also detect muscle and glial fibers	Less common	[36,40]
**Carstair’s Method **	Platelets and fibrin	Can also detect RBCs, collagen, and muscle	Uncommon	[57]
**Immunohistochemical analysis**	vWF (ab6994) [39,46,47,57,60] Neutrophils (CD66b) [46,63] Neutrophil elastase [58,63,64] Myeloperoxidase [64] WBCs (CD45) [36,47,64] T cells (CD3) [35,59,64] B cells (CD20) [35,59,64] Monocytes (CD14) [64] Neutrophils, eosinophils, monocytes (CD 15) [57] Endothelial cells (CD34 [38]) NETs (H3Cit) [46,63] Platelet glycoprotein Ib (CD42b) [32,60] Ki-M1P, macrophages, and activated platelets (CD68) [35,36,59] Platelets (CD31) [40,57] Platelet activation (CD42b) [46] Platelet glycoprotein IIb/IIIa (CD61) [37], (CD41) [64] RBC (Glycophorin A) [40,46] Fibrinogen antibodies [46,64] Fibrin [46] PAI-1 [46] PN-1 [46]	Can provide more detail than histological staining More precise and quantitative	Costly Requires well-trained histopathologists	As per the second column

Abbreviations: *RBC* red blood cell; *WBC* white blood cell; *LAA* large artery atherosclerosis; *VWF* von Willebrand factor; *NET* neutrophil extracellular trap; *PAI-1* plasminogen activator inhibitor-1; *PN-1* protease nexin-1.

**Table 2 ijms-23-12373-t002:** Associations between thrombus components and stroke treatment success, etiology, imaging, and clinical outcomes.

Clot Component	Treatment Success	Etiology	Imaging Markers	Clinical Outcome/Severity/Prognosis
**RBC**	Associated with favorable reperfusion outcomes [41,44,50,53] No significant association [32,37,38,56]	Associated with LAA or non-cardioembolic origin [32,34,40,44] Associated with the cardioembolic origin [37,50] No significant association [38,51]	Associated with higher attenuation on NCCT or HMCAS [39,40,42,44,50,54] Associated with positive SVS [37,53] No significant association [32,65] Associated with increased perviousness [55]	No significant association [60]
**Polyhedrocytes**	Associated with worse reperfusion outcome [66]	None found	None found	Associated with higher NIHSS score [66]
**Platelets/Fibrin**	Associated with worse reperfusion outcomes [35,60] No significant association [32,37,38,56]	Associated with cardioembolic origin [32,34,35,49] Associated with LAA or non-cardioembolic origin [50] No significant association [38,40] Platelets (without fibrin) higher in LAA origin [56]	Associated with negative SVS [37] Associated with isodense clots on NCCT [51] Associated with decreased perviousness [39,55] Associated with increased perviousness [49,67]	Higher chance of distal embolism [35] No significant association [60]
**WBCs**	Associated with worse reperfusion outcomes [42]	Associated with cardioembolic origin [34,35,42]	Associated with decreased perviousness [55]	Associated with higher NIHSS score at discharge [42] Associated with higher mRS score at 90 days [42]
**NETs**	Associated with worse reperfusion outcomes [63,68]	Associated with the cardioembolic origin [63]	None found	Associated with higher NIHSS score at discharge [64] Associated with higher mRS score at 90 days [64]
**VWF**	Associated with worse reperfusion outcomes [57]	None found	None found	Associated with higher pre-intervention NIHSS score [57]
**Extracellular DNA**	None found	Associated with the cardioembolic origin [69]	None found	None found
**Endothelial Cells**	None found	None found	None found	None found
**Bacteria**	Associated with worse reperfusion outcomes [43]	Associated with infectious pathologies [7,43]	None found	None found
**Calcifications**	Associated with worse reperfusion outcomes [48,70]	None found	Associated with rounder and more hyperdense clots on imaging [71]	Associated with higher mortality and recurrent stroke [72]
**Other**	Vascular wall components are associated with increased thrombectomy passes [52]	None found	None found	None found

Abbreviations: *RBC* red blood cell; *LAA* large artery atherosclerosis; *NCCT* non-contrast computed tomography; *HMCAS* hyperdense middle cerebral artery sign; *SVS* susceptibility vessel sign; *NIHSS* National Institutes of Health Stroke Scales; *WBC* white blood cell; *mRS* modified Rankin Scale; *NET* neutrophil extracellular trap; *VWF* von Willebrand factor.

## Data Availability

The original contributions presented in the study are included in the article, further inquiries can be directed to the corresponding author.

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
