# Peer review of "Clot Morphology in Acute Ischemic Stroke Decision Making"

_ijms, 2022, doi:10.3390/ijms232012373_

Round 1
Reviewer 1 Report
Scrupulously and very well written review about the possibility of analyzing the structure of a clot as a potential element of diagnosis and prognosis in ischemic stroke. I have no critical remarks, the paper is painstakingly written and it concerns an issue not commonly described in the literature. I am only asking the authors to discuss the possibility of evaluating EVs in a thrombus. It is an interesting concept that EVs with procoagulant properties may play a key role in the pathophysiology of stroke (doi: 10.1111/ene.12591), and it would be interesting to discuss the potential role of EVs from an isolated thrombus.
Author Response
We thank the reviewer for the review of our work. We have made the best of our efforts to address various points/comments made by the reviewer. Our point-by-point rebuttal to comments are provided below. A revised version of the manuscript with track changes are attached for your perusal.
C#1: Scrupulously and very well written review about the possibility of analyzing the structure of a clot as a potential element of diagnosis and prognosis in ischemic stroke. I have no critical remarks, the paper is painstakingly written, and it concerns an issue not commonly described in the literature. I am only asking the authors to discuss the possibility of evaluating EVs in a thrombus. It is an interesting concept that EVs with procoagulant properties may play a key role in the pathophysiology of stroke (doi: 10.1111/ene.12591), and it would be interesting to discuss the potential role of EVs from an isolated thrombus.
Reply# Thank you for the positive review of our work and the suggestion on EVs. We have included a paragraph discussing the role of EVs in stroke as well as potential to evaluate EVs from isolated thrombus.
Page 8 (Lines 214-219)
Interestingly, extracellular vesicles (EVs) derived from blood may be of potential utility in stroke etiology delineation and indicating the state of cerebrovascular disease (1). EVs can mediate coagulation through anticoagulant or fibrinolytic pathways in addition to their procoagulant capabilities (2, 3). In this context, evaluation of EVs in isolated thrombus and their putative role in differentiating stroke types offer another avenue for future research (3).

Reviewer 2 Report
Overall this review paper provides an adequate appraisal of the current state of the art in the analysis of acute ischemic stroke thrombi via histological means. The review will provide those that are interested and not expert in the field with a top level synopsis of the current research in the area.
The review doesn't add any new insights or provides strong recommendation for future research focus.
There are some minor issues with the manuscript that have been highlighted through comments in the attached pdf document.

Author Response
We thank the reviewer for the review of our work. We have made the best of our efforts to address various points/comments made by the reviewer. Our point-by-point rebuttal to comments are provided below. A revised version of the manuscript with track changes are attached for your perusal.
C#1: Overall this review paper provides an adequate appraisal of the current state of the art in the analysis of acute ischemic stroke thrombi via histological means. The review will provide those that are interested and not expert in the field with a top level synopsis of the current research in the area. The review doesn't add any new insights or provides strong recommendation for future research focus. There are some minor issues with the manuscript that have been highlighted through comments in the attached pdf document.
Reply: We thank the reviewer for the review of our work and positive review and consideration.
C#2: schematic could be improved to show realistic scale of aspiration catheter with typical vessel size treated, e.g 0.060" or 0.070" ID catheter in a 2.5-3.0mm M1 segment of iCA. Also the baloon on the baloon guide does not touch the vessel walls, which is essential to how it functions
Reply: Thank you for the observations. The schematic has been updated and the legend has been expanded to incorporate the suggestions made by the reviewer, as below.
Page 2
Figure 1. Various techniques used in endovascular thrombectomy. A schematic displaying four techniques. a) describes the stent-retriever technique in which an aspiration catheter (e.g., a catheter with an inner diameter (ID) of 0.060" or 0.070") is inserted through the femoral or radial artery and guided up to the specific vessel (e.g., approximately 2.5-3.0 mm in the M1 segment of the middle cerebral artery (MCA)) blocked by the clot. The guidewire and microcatheter cross the clot, and the stent is unsheathed and integrated with the clot, providing immediate reperfusion in most cases. Then, the stent and microcatheter are retrieved, ideally along with the clot [27]. b) Aspiration catheters, which are larger than a microcatheter but not a fully deployed stent retriever, aspirate the clot in order to remove it from circulation [28]. c) The Solumbra technique utilizes both aforementioned techniques, simultaneously aspirating and using a stent to retrieve the clot. d) A balloon guide catheter can be inflated in place in the vessel, stopping antegrade blood flow and causing brief retrograde flow, diminishing the risk of distal embolization.
C#3: This is mostly the case, but not always.
Reply: We have included this as below
“…providing immediate reperfusion in most cases.”
C#4: larger than what? they are larger than a m/c but not a fully deployed SR.
Reply# This has been revised accordingly, as below
Aspiration catheters, which are larger than a microcatheter but not a fully deployed stent retriever, aspirate the clot in order to remove it from circulation
C#5: how does the clot influence stroke etiology? surely it is the etiology that influences the clot which in turn causes the stroke?
Reply# We have revised the sentence as below
“An overview of these components will be discussed, followed by a critical evaluation of the evidence regarding the association of clot features with stroke etiology, imaging, management, and prognosis, all of which are summarized in Table 1."
C#6: How definitive is this statement? MSB is not specific for platelets.
Reply# MSB stains can identify platelet rich clots. We have added reference to support this (see (6))
C#7: no mention of paradoxical embolism through PFO in heart
Reply# Thank you for suggestion and as indicated we have added following paragraph to Page 9.
Patent foramen ovale (PFO)-induced paradoxical embolism is likely the most common cause of cryptogenic stroke [92]. A PFO may be presented in up to 30% of the population. Individuals with a PFO have a 0.1% and 1% annual risk of cryptogenic and recurrent strokes, respectively [93].
C#8: The resolution of imaging modalities to accurately measure hyperdensity/svs should be discussed in the context of the size of clots/vessels being imaged and through the skull.
Reply# Following sentences have been added to Pages 9-10.
The resolution of imaging modalities to accurately measure HMCAS and SVS may depend on the size of clots/vessels being imaged and the thickness of the multiplanar imaging of the skull. Thin-section multiplanar imaging should be preferred [100].

Reviewer 3 Report
In the manuscript “Clot Morphology in Acute Ischemic Stroke Decision Making” by Huang and Bhaskar. A review explaining the role of clot morphology in ischemic stroke. It is good review explaining clot components from stroke patients and its role in stoke pathophysiology. I have following concerns.
1. Currently role of neutrophil/NETs is hot topic in stroke research. Although authors have discussed NETs and VWF, authors should discuss role of NETS in stroke pathology with recent research (HMGB1, PKM2 and integrin) (e.g.: J Clin Invest. 2022;132(10)., Blood. 2022;139(8):1234-1245) .
A. What is the main question addressed by the research? Is it relevant and interesting?:
Ans. A review explaining the role of clot morphology in ischemic stroke, it is relevant and interesting for understanding pathology for stroke.
B. How original is the topic? What does it add to the subject area compared with other published material?
Ans. It is new topic as an in-depth understanding of clot morphology will help to design treatment for stroke.
C. Is the paper well written? Is the text clear and easy to read?:
Ans. It is well written and test is easily readable and understable.
D. Are the conclusions consistent with the evidence and arguments presented? Do they address the main question posed?
Ans. They have concluded very well.
Author Response
We thank the reviewer for the review of our work. We have made the best of our efforts to address various points/comments made by the reviewer. Our point-by-point rebuttal to comments are provided below. A revised version of the manuscript with track changes are attached for your perusal.
C#1: In the manuscript “Clot Morphology in Acute Ischemic Stroke Decision Making” by Huang and Bhaskar. A review explaining the role of clot morphology in ischemic stroke. It is good review explaining clot components from stroke patients and its role in stoke pathophysiology. I have following concerns.
Reply: We thank the reviewer for the review of our work and positive review and consideration.
C#2: 1. Currently role of neutrophil/NETs is hot topic in stroke research. Although authors have discussed NETs and VWF, authors should discuss role of NETS in stroke pathology with recent research (HMGB1, PKM2 and integrin) (e.g.: J Clin Invest. 2022;132(10)., Blood. 2022;139(8):1234-1245) .
Reply# Thank you for the suggestion. We agree this is an important area of future research. Following paragraph has now been added.
"Recent research has also implicated a pathological role of NETs in ischemic stroke indicating the therapeutic potential of NET-inhibitory factor (nNIF) in stroke [87,88]. A study by Denorme et al reported NET-forming neutrophils throughout the brain tissue of patients who had ischemic strokes [87]. The authors also found that increased plasma NET biomarkers were associated with worse stroke outcomes. Furthermore, stroke patients had elevated plasma and platelet surface–expressed high-mobility group box 1 (HMGB1). Another study by Dhanesha et al found that the nuclear pyruvate kinase muscle 2 (PKM2), a regulator of systemic inflammation, was upregulated in neutrophils following the onset of ischemic stroke in both humans and mice [88]. Neutrophil hyperactivation was mediated by PKM2. By reducing postischemic cerebral thrombo-inflammation, PKM2 deficiency in myeloid cells improved both the short- and long-term outcomes of stroke [88]."
C#3: A. What is the main question addressed by the research? Is it relevant and interesting?: Ans. A review explaining the role of clot morphology in ischemic stroke, it is relevant and interesting for understanding the pathology for stroke.
Reply# We thank the reviewer for a positive review of our work.
C#4: B. How original is the topic? What does it add to the subject area compared with other published material? Ans. It is new topic as an in-depth understanding of clot morphology will help to design treatment for stroke.
Reply# Thank you for the positive review of our work.
C#5: C. Is the paper well written? Is the text clear and easy to read?: Ans. It is well written and test is easily readable and understable.
Reply# We thank the reviewer for a positive review of our work.
C#6: D. Are the conclusions consistent with the evidence and arguments presented? Do they address the main question posed? Ans. They have concluded very well.
Reply# We thank the reviewer for a positive review of our work.

Reviewer 4 Report
Dear Authors,
Paper Huang and Bhaskar entitled “Clot Morphology in Acute Ischemic Stroke Decision Making” is devoted to substantial review of thromboembolism pathogenesis, clot composition, etiology and imaging. The significance of the "clot" factor in clinical outcomes and treatment of ischemic stroke events is also discussed. The review also indicates the limitations and prospects (future direction) on this matter.
The paper Huang and Bhaskar is timely and relevant. All Tables and Figures are informative and well structured. Thus, the paper is pleasant and interesting to read.
There are minor comments
1. Recently, several reviews on related topics "Clot in Ischemic Stroke" have appeared (PMID: 34649378, 34279484, 34528378, 34045301, 33341247, etc). The authors should take them into account in systematizing the data and formulating the conclusion on their review.
2. The authors should provide the soft tool with which the images on the figures were created.
3. The authors used a mixture of British and American English. For example, "ischemic" (line 2) is American English and "ischaemic" (line 32) is British English. A uniform language style should be observed and appropriate corrections should be made throughout the text.
Thus, paper Huang and Bhaskar can be appropriate for publication in IJMS after implementation these comments.
Sincerely,
Referee
Author Response
We thank the reviewer for the review of our work. We have made the best of our efforts to address various points/comments made by the reviewer. Our point-by-point rebuttal to comments are provided below. A revised version of the manuscript with track changes are attached for your perusal.
C#1: Dear Authors,
Paper Huang and Bhaskar entitled “Clot Morphology in Acute Ischemic Stroke Decision Making” is devoted to substantial review of thromboembolism pathogenesis, clot composition, etiology and imaging. The significance of the "clot" factor in clinical outcomes and treatment of ischemic stroke events is also discussed. The review also indicates the limitations and prospects (future direction) on this matter.
The paper by Huang and Bhaskar is timely and relevant. All Tables and Figures are informative and well-structured. Thus, the paper is pleasant and interesting to read.
Reply: We thank the reviewer for the review of our work and the positive review and consideration.
There are minor comments
C#2: 1. Recently, several reviews on related topics "Clot in Ischemic Stroke" have appeared (PMID, 34279484, 34528378, 34045301, 33341247, etc). The authors should take them into account in systematizing the data and formulating the conclusion on their review.
Reply# We thank the reviewer for the comment. We have added the following sentences to the Introduction (Page 1: Lines 43-49)
Whilst qualitative assessment of the embolus is still crucial in the diagnosis of individual cases, especially when identifying atypical causes of stroke; there is yet no definitive clot signature for the precise diagnosis of the etiology of stroke [14-16].
C#3: 2. The authors should provide the soft tool with which the images on the figures were created.
Reply# The images were generated using several tools including Powerpoint, Canva and other soft tools.
C#4: 3. The authors used a mixture of British and American English. For example, "ischemic" (line 2) is American English and "ischaemic" (line 32) is British English. A uniform language style should be observed and appropriate corrections should be made throughout the text.
Reply# We have corrected this throughout. We thank the reviewer for pointing this out.
C#5: Thus, paper Huang and Bhaskar can be appropriate for publication in IJMS after implementation these comments.
Reply: We thank the reviewer for the review of our work and positive review and consideration.
